# Distinct multiple fermionic states in a single topological metal

M. Mofazzel Hosen[1], Klauss Dimitri[1], Ashis K. Nandy[2], Alex Aperis [2], Raman Sankar [3,4], Gyanendra Dhakal[1], Pablo Maldonado [2], Firoza Kabir[1], Christopher Sims[1], Fangcheng Chou[3], Dariusz Kaczorowski[5], Tomasz Durakiewicz [6], Peter M. Oppeneer [2] & Madhab Neupane[1]

Among the quantum materials that have recently gained interest are the topological insulators, wherein symmetry-protected surface states cross in reciprocal space, and the Dirac nodal-line semimetals, where bulk bands touch along a line in k-space. However, the existence of multiple fermion phases in a single material has not been verified yet. Using angle-resolved photoemission spectroscopy (ARPES) and first-principles electronic structure calculations, we systematically study the metallic material $Hf_2Te_2P$ and discover properties, which are unique in a single topological quantum material. We experimentally observe weak topological insulator surface states and our calculations suggest additional strong topological insulator surface states. Our first-principles calculations reveal a one-dimensional Dirac crossing—the surface Dirac-node arc—along a high-symmetry direction which is confirmed by our ARPES measurements. This novel state originates from the surface bands of a weak topological insulator and is therefore distinct from the well-known Fermi arcs in semimetals.

[1] Department of Physics, University of Central Florida, Orlando, FL 32816, USA. [2] Department of Physics and Astronomy, Uppsala University, P. O. Box 516S-75120 Uppsala, Sweden. [3] Center for Condensed Matter Sciences, National Taiwan University, Taipei 10617, Taiwan Institute of Physics, Academia Sinica, Taipei 10617, Taiwan. [4] Institute of Physics, Academia Sinica, Taipei 10617, Taiwan. [5] Institute of Low Temperature and Structure Research, Polish Academy of Sciences, 50-950 Wroclaw, Poland. [6] Condensed Matter and Magnet Science Group, Los Alamos National Laboratory, Los Alamos, NM 87545, USA. Correspondence and requests for materials should be addressed to A.K.N. (email: ashis.nandy@physics.uu.se) or to A.A. (email: alex.aperis@physics.uu.se) or to M.N. (email: Madhab.Neupane@ucf.edu)

The discovery of topological insulators (TIs)[1–5] has invigorated intense research efforts in the quest for novel nontrivial surface states (SSs) in a wider group of quantum materials[6–10], which includes bulk insulators, semimetals, and metals. These SSs can manifest themselves from multiple origins such as strong spin-orbit coupling (SOC), accidental band touching, symmetry protection, or strong electronic correlation effects[5,11–14]. Uncovering the origin of topological protection of the surface states has hence become an important research topic.

A tetradymite-type Bi-based compound was the first experimentally discovered TIs in which the surface state was realized through the SOC effect and protected by time-reversal symmetry (TRS)[1,2]. Moving from topological surface states (TSSs) in the TIs, topological Dirac semimetals (TDSs) such as $Cd_3As_2$[6,7,15] and $Na_3Bi$[16–19] were discovered to have band touchings between the valence and conduction bands at certain discrete $k$-points in the Brillouin zone (BZ) and naturally host linear Dirac dispersions in three-dimensional (3D) momentum space. These TDSs with fourfold degeneracy are protected by additional crystalline symmetries other than the TRS and inversion symmetry[6,16,17,20]. Another subsequently discovered semimetallic class, that of Weyl semimetals, is featured with pairs of bulk Dirac-cones and the connecting Fermi-arc SS[10,21–23]. The metallic surface states of the Weyl semimetals show photon-like linear band dispersion and exhibit a variety of exotic properties that include large magnetoresistance and high carrier mobility[10,21–23]. Evolving from the Weyl semimetals, nodal-line semimetals (an extended TDS class) possess a one-dimensional (1D) bulk band touching in a loop in momentum space and are accompanied by so-called drumhead surface bands[24–27]. Here, one needs additional crystalline symmetries to protect the degeneracy of the line-node against perturbation, such as mirror symmetry[25], or nonsymmorphic symmetry[26–28]. The high density of states in the topological two-dimensional (2D) fermion state provides a possible route to discover materials that could exhibit high-temperature surface superconductivity[29].

In spite of the already discovered materials hosting one of the mentioned quantum phases, materials that exhibit several TSSs at distinct reciprocal space locations would be highly desirable as these could unlock an interplay of topological quantum phenomena. Here, through the use of ARPES and ab initio calculations, we report the discovery of topologically distinct fermionic SSs, multiple Dirac crossings, and the Dirac-node arc[30], in a single topological metal $Hf_2Te_2P$, a compound from the 221 family. This family of materials has recently drawn interest; replacing Hf by a lighter atom Zr, one obtains $Zr_2Te_2P$ which is a strong topological metal with multiple Dirac cones[31]. Unlike the closed loop line-nodes accompanied with drumhead surface states, here in $Hf_2Te_2P$, the surface band crossing, resembling the Dirac state in graphene, is extended to a short line along a high-symmetry direction—a Dirac node arc. Our Density Functional Theory (DFT) calculations predict the presence of weak topological states below the Fermi level, strong topological states above the Fermi level and the surface Dirac-node arc below the Fermi level. Our ARPES measurements demonstrate the existence of the weak topological Dirac states and the Dirac-node arc in a single topological metal $Hf_2Te_2P$. Our measurements furthermore reveal the presence of multiple pockets at the Fermi level ($E_F$) collectively comprising of a sixfold flower with a petal-shaped Fermi surface (FS). Apart from several Dirac cones located at the $\Gamma$ point at different binding energies ($E_B$) below and above the Fermi level, our most important observation is that of the Dirac-node arc seen here in a topological metal, centered at the M point along the $\Gamma$–M–$\Gamma$ direction in the energy momentum plane. In contrast to the Dirac-node arc of unambiguous origin observed in topological line-node semimetals[32], our DFT study reveals that here it

is a signature of weak topological $\mathbb{Z}_2$ invariants and protected by in-plane time-reversal invariance[33]. Consequently, this material could be the first system to realize the coexistence of both weak and strong $\mathbb{Z}_2$ invariants due to the presence of multiple bulk topological bands.

## Results

**Crystal structure and sample characterization**. The materials of the 221 family are of particular interest due to their tetradymite-type layered crystal structure, mostly found in typical 3D TIs as e.g., $Bi_2Te_3$, $Bi_2Se_3$, and $Sb_2Te_3$[5,31,34–36], having a threefold rotation symmetry about the $z$-axis. Similar to the well-known TI $Bi_2Te_2Se$[37], $Hf_2Te_2P$ crystallizes in a rhombohedral crystal structure with space group $R\bar{3}m$ (No. 166)[38]. The conventional (hexagonal) unit cell consists of three basic quintuple layers (QL), each with stacking sequence Te-Hf-P-Hf-Te (see Fig. 1(a)). The atoms within a QL are covalently bonded, whereas between layers they are bonded by weak Van der Waals forces which facilitate to cleave along the {001} basal plane, as shown in Fig. 1(a). The primitive (rhombohedral) unit cell possesses an inversion symmetry depicted by the red star. The crystal also respects the TRS as well as preserves the reflection symmetry with respect to the mirror planes {110}, {010}, and {100} (pertaining to the conventional unit cell). This reflection symmetry may play a distinct role in conserving in-plane time-reversal invariance and hence, protecting the Dirac-node arc, as discussed later. Figure 1(b) shows the temperature dependence of the electrical resistivity of $Hf_2Te_2P$ which indicates the metallic nature of this material. The transverse magnetoresistance $MR = [\rho(T, H) - \rho(T, 0)]/\rho(T, 0)$ attains a value of about 100% below 10 K. At higher temperatures, the magnitude of MR rapidly decreases and drops below 1% above 150 K. The bulk and (111)-projected surface BZs of the primitive unit cell are shown in Fig. 1(d). The high symmetry points $\Gamma$, Z, F, and L are on the mirror planes of either {110} or {100} type in the bulk BZ. The projected surface BZ is perpendicular to the mirror planes. The center of the surface BZ is denoted as the $\Gamma$ point, the K points are at the corners of the surface BZ, and M is the mid-point of two adjacent corners. Figure 1(e) shows the spectroscopic core level measurements of $Hf_2Te_2P$. We observe sharp peaks of Hf $4f$ at around 14.2 eV and 15.9 eV, and Te $4d$ at around 40 eV. This indicates that the sample used in our measurements is of good quality.

**FS and Dirac node arc**. We now report the experimental results that reveal the details of the electronic structure including the FS and the TSSs with multiple Dirac-like features. We probed the electronic structure of $Hf_2Te_2P$ through the use of ARPES with low incident photon energy (80–100 eV). Figure 2(a) shows the FS maps with ARPES intensity integrated in a 20-meV energy window for $Hf_2Te_2P$ at various photon energies. It reveals the presence of multiple FS pockets such as the petal-shaped six electron-like pockets along the $\Gamma$-M directions and a hole-like pocket at the zone center $\Gamma$, resembling a sixfold flower-like FS. This type of pronounced electronic dispersion is extremely rare in Dirac-type materials (see also Supplementary Fig. 1; Supplementary Note 1). The sixfold flower petal-like FS reflects the threefold rotational and inversion symmetry of the lattice.

Figure 2(b–d) shows the energy dispersion maps along various key directions indicated on the FS of Fig. 2(a) (see also Supplementary Figs. 2–4; Supplementary Notes 2 and 3). Figure 2 (b, c) shows the dispersion maps along the K–$\Gamma$–K and M–$\Gamma$–M directions, respectively, at different photon energies as denoted on the plots. A clear linearly dispersive state (Dirac-like) is observed around the $\Gamma$ point with a wider range of band dispersion. The chemical potential of the Dirac crossing is found

at the top of this highly linearly dispersive state. Figure 2(c) shows the dispersion maps along the Γ–M direction. These linearly dispersive maps are found to be unchanged with respect to the varying photon energy, supporting the 2D nature of this state (see also Supplementary Figs. 5 and 6; Supplementary Note 3). Furthermore, we carefully analyze the dispersion map along the K–Γ–K and at $E_B \approx 1.2$ eV, where we observe a second Dirac cone-like feature. However, the Dirac cone is not clearly observed due to the broad ARPES intensity; therefore, we confirmed it by taking a second-derivative curvature plot and energy distribution curves (EDCs) of this dispersion map (see Supplementary Fig. 2; Supplementary Note 2). Rather, in Fig. 2(d), an extremely sharp Dirac-like linear dispersion within a wide energy window is observed along the K–M–K direction for all three photon energies, and therefore more likely to be due to the SSs (see also Supplementary Figs. 2, 3, and 7; Supplementary Notes 2 and 3). Importantly, the linearly dispersive energy range is as large as 2.2 eV below and above the Fermi level which is even larger than that of the recently reported nodal-line semimetal ZrSiS[26,27]. Finally, the Dirac-like band touching is observed at the M point at $E_B \approx 1.1$ eV. Importantly, our ARPES results establish the presence of multiple Dirac surface features (two crossings) located at different binding energies at different high-symmetry points Γ and M. The Dirac-like feature close to the M point is much broader than the Dirac-cone around the zone center Γ.

Now we report our most important ARPES observation, guided by the theoretical calculations, of the Dirac-node arc, as a signature of weak TI. Figure 3(a) shows the constant energy contours at a photon energy of 100 eV as a function of binding energy $E_B$, as noted on the plots. Moving towards higher $E_B$, we observe that the ellipsoidal petal-shaped features are gradually shrinking in width along the K–M–K direction whereas the small circular pocket at the Γ point is increasing in size. This indicates the hole-like nature of the pocket at the zone center and the electron-like nature of the ellipsoidal pockets. At $E_B \approx 1$ eV, we

observe that the ellipsoidal-like pocket disappears and finally forms a 1D line-like feature along the high-symmetry M–Γ–M direction. A closer look in energy, Fig. 3(b), shows that the novel line-like fermionic state begins to flatten at a binding energy of 1.1 eV. Next, we focus on the energy dispersion around the M point in more detail according to the cuts shown in the momentum dispersion curve at $E_B = 1$ eV, Fig. 3(b). Along cuts No. 1 to No. 9, the band dispersions shown in Fig. 3(c) describe the detailed evolution of the Dirac features along the K–M–K direction. At the M point in cut No. 5, a very sharp linear dispersion starts to cross at $E_B = 1.1$ eV, see also Fig. 2(d). The Dirac crossing moves up in energy on both sides of the M point, making cuts No. 2–4 and cuts No. 6–8 symmetrical. Importantly, each crossing resembles the Dirac state in graphene. Therefore, in the proximity of TRS point M the movement of such Dirac crossing forms a line, the Dirac-node arc in energy-momentum space. Moving further away from the M point, cuts No. 1 and 9 show a slightly gapped Dirac cone with a largely linear dispersion and the bulk states dominate over the surface contribution (see Supplementary Figs 8 and 9; Supplementary Note 4).

**Bulk and surface states.** For a detailed understanding of the novel ARPES features, we now focus on the electronic structure calculations of $Hf_2Te_2P$ based on DFT. All the observed features are well explained by our ab initio calculations which guided us to identify the surface Dirac-node arc, and disentangle bulk and surface origins. Figure 4(a) shows the bulk electronic structure of $Hf_2Te_2P$ along the high-symmetry directions including four time-reversal-invariant momenta (TRIM), calculated with SOC. Several bands cross the Fermi energy that are formed of hybridized Hf-$d$ (blue circles) and Te-$p$ (red circles) states; the size of the circles indicates the contribution for a specific orbital. The bands close to $E_F$ around the F point are almost flat bands, i.e., they are approximately dispersionless and for the sake of simplicity, these

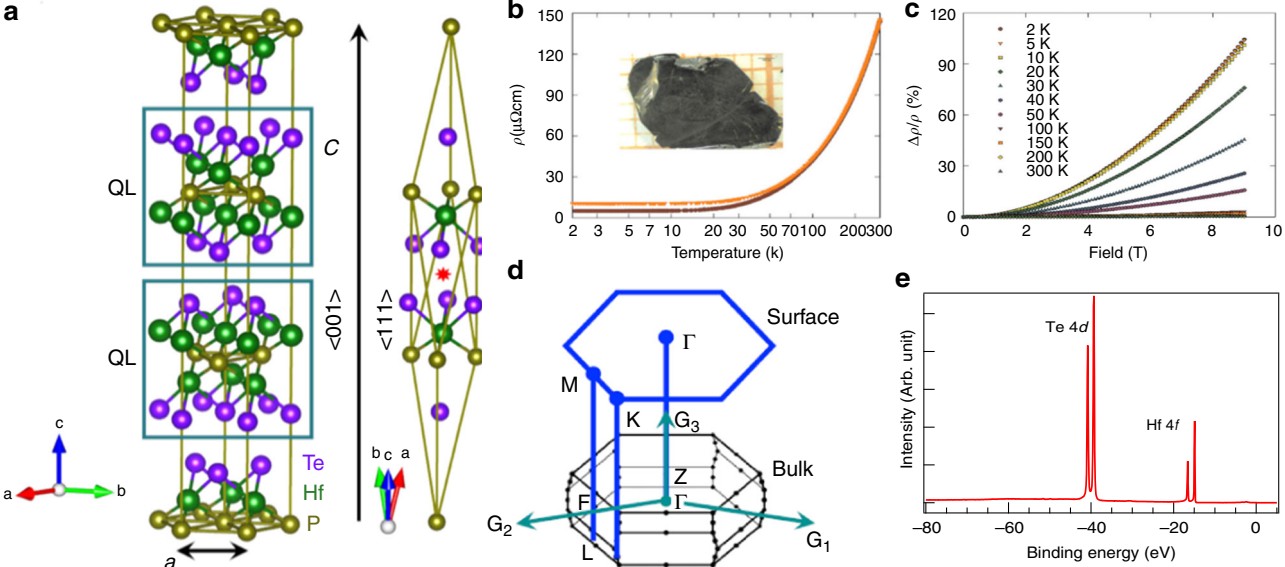

**Fig. 1** Crystal structure and sample characterization of $Hf_2Te_2P$. **a** The rhombohedral tetradymite crystal structure of $Hf_2Te_2P$, depicted in the conventional (hexagonal) and primitive (rhombohedral) unit cells. The stacking of quintuple layers (QL) is depicted by the blue square. The inversion center is indicated with the red star. **b** Temperature dependence of the electrical resistivity measured on a single crystal of $Hf_2Te_2P$ in a zero (brown circles) and 9 T (orange triangles) magnetic field applied perpendicular to the current flowing within the basal plane of the crystallographic unit cell. The inset shows a picture of one of the single crystal grown for the present research. **c** Magnetic field dependencies of the transverse magnetoresistance measured for a range of different temperatures on a single crystal of $Hf_2Te_2P$ with current flowing within the basal plane of the crystallographic unit cell. **d** 3D bulk Brillouin zone and its projection on the hexagonal surface Brillouin zone of the $Hf_2Te_2P$-crystal. High symmetry points are marked in the plot. **e** Core level spectroscopic measurement of $Hf_2Te_2P$. Sharp peaks of Te 4$d$ and Hf 4$f$ are observed which indicate good sample quality

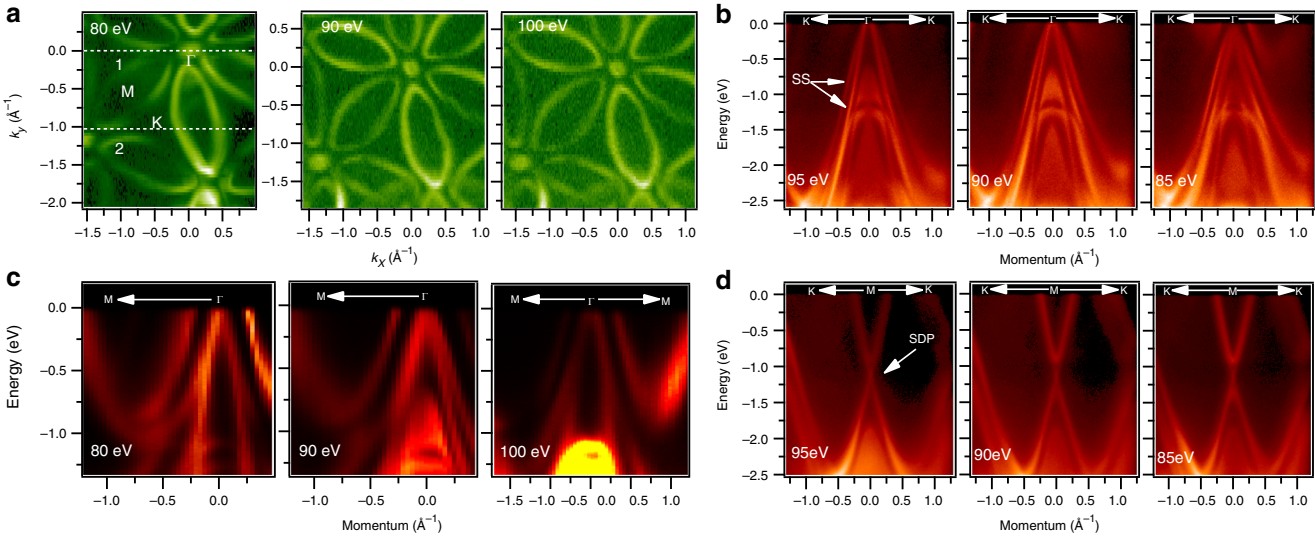

**Fig. 2** Fermi surface and observation of multiple fermionic states. **a** Fermi surface maps at various photon energies. Photon energies are marked on the plots. The white dashed lines marking No. 1 and 2 denote the direction of the dispersion maps. **b**–**d** Dispersion maps measured along various high-symmetry directions for different photon energies. These data were collected at the SIS-HRPES end station at the SLS, PSI, at a temperature of 18 K

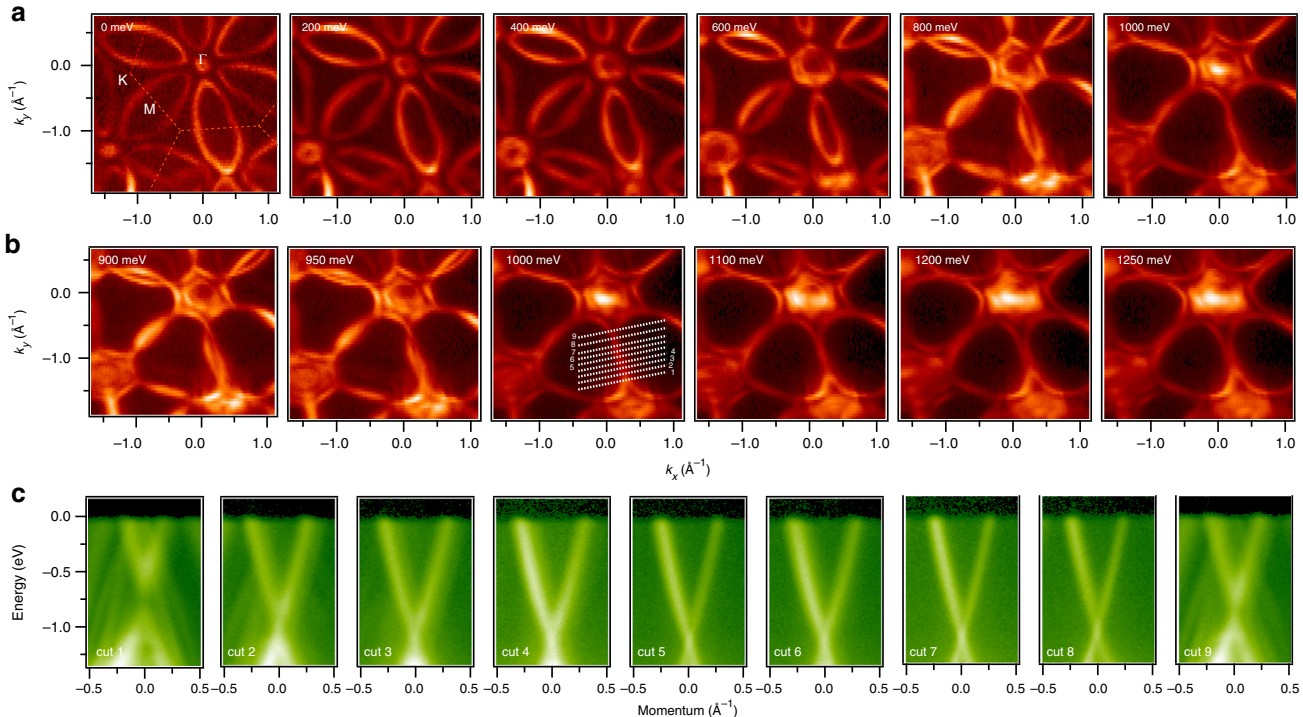

**Fig. 3** Experimental observation of the Dirac node arc. **a** Constant energy contour plots at various binding energies. **b** Constant energy contour plots closer to the Dirac-node arc. Binding energies are given in the plots. **c** Dispersion map along the K–M–K direction along the cut directions indicated in the 1000-meV constant energy contour panel of **b**. All data were collected at the SIS-HRPES end station at the SLS, PSI at a photon energy of 100 eV with a temperature of 18 K

are labeled as bands A, B, and C which are predominantly formed of Te-$p$, Te-$p$, and Hf-$d$ orbital characters, respectively. Irrespective of the $E_F$ position, direct energy gaps exist between the band sets (A, B) and (B, C) in the whole BZ, owing to the SOC; this direct energy gap determines the probable topology of SSs. Apart from a clear band inversion of the $d$–$p$ type between bands B and C at the $\Gamma$ point, there is an additional inversion among the topmost valence bands at the L point induced by SOC, highlighted by the green boxes. These band inversions are usually considered as the origin of the topological nontriviality i.e.,

nontrivial TSSs. As shown in Fig. 1(a), the primitive unit cell with the inversion center at the red star further allows us to determine the parity of the Bloch wave function which is consistent with that of the corresponding atomic orbital: + for the $s$ and $d$ orbitals and − for the $p$ orbital.

To confirm the topological origin of the observed features in the ARPES experiment, we calculate the four topological $\mathbb{Z}_2$ invariants, $(\nu_0; \nu_1, \nu_2, \nu_3)$, assuming $E_F$ can be tuned into the pseudogap between band sets (A, B) and (B, C). This calculation is based on considering the parity of the predominant orbital

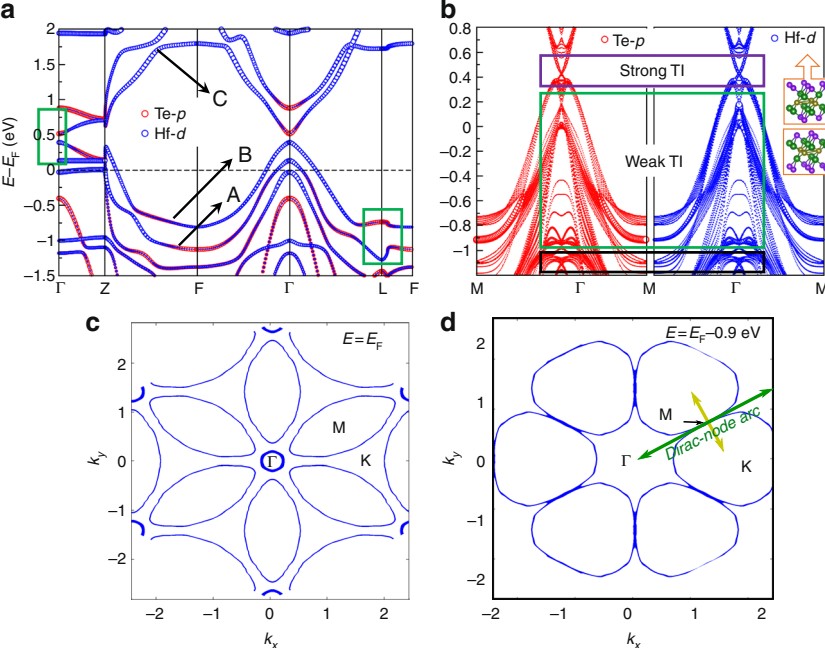

**Fig. 4** Calculated multiple fermionic states. **a** The bulk electronic structure, calculated with spin-orbit coupling, along high-symmetry directions. Blue and red circles indicate the Hf-d and Te-p character of the band states, respectively. Band inversions are highlighted by the green rectangles. **b** Calculated surface states for the (111) surface with Hf and Te character indicated. The purple rectangle highlights the strong TI surface state stemming from bulk bands B and C whereas the green rectangle highlights the weak TI surface states due to bulk bands A and B. An additional gapless surface state is highlighted with a black box. **c**, **d** Calculated Fermi surface and constant energy contour 900 meV below the Fermi level, respectively. At 900 meV below the Fermi level, we observe a symmetry protected surface Dirac-node arc, denoted by the green arrow

character at the four irreducible TRIM points of the BZ, as described by Fu and Kane[39]. Interestingly, within the band set (A, B), we find $\nu_0 = 0$, but the other topological invariants are calculated to be nonzero, $(\nu_1, \nu_2, \nu_3) = (1, 1, 1)$—a weak topological $\mathbb{Z}_2$ invariant. Once we consider the pseudogap between bands B and C, strikingly, $\nu_0$ changes from 0 to 1 due to the band inversion at $\Gamma$, while weak invariants $(\nu_1, \nu_2, \nu_3)$ remain unaltered—a strong topological $\mathbb{Z}_2$ invariant. Therefore, $Hf_2Te_2P$ is a unique quantum material carrying multiple topological bands along with both weak and strong $\mathbb{Z}_2$ invariants. Both $\mathbb{Z}_2$ topological invariants guarantee the existence of TSSs on the (111) surface.

Indeed, in our Te terminated (111) surface calculations, we observe multiple topologically protected Dirac states, see Fig. 4(b). First of all, a pair of linearly dispersive TSSs exists around the $\Gamma$ point inside the $d-p$ inversion gap (between bulk band set (B, C)) with the Dirac point lying at about 0.42 eV above the $E_F$. At about 0.17 eV above $E_F$, another Dirac point is observed at the same $\Gamma$ point protected by the band topology of bulk states A and B. Here, the surface character of the Dirac dispersion is strongly mixed with the bulk character. Unfortunately, the above mentioned two Dirac points cannot be visualized in our ARPES measurements as only occupied electronic bands can be probed by this technique. However, our constant energy contour plots in Fig. 4(c, d) confirm that the first Dirac point is derived from the sharp corner of the six petal-shaped pockets meeting at a point above $E_F$ while the later one is derived from the circular ring at the zone center as observed in ARPES, see Fig. 2(a). In stark contrast to a single Dirac point at 0.42 eV above $E_F$, a nontrivial line-like feature is observed around $E_B \sim 0.9$ eV at the M point along the $\Gamma$–M–$\Gamma$ direction originating from the same band topology of the bulk band set (A, B). Such highly anisotropic bands around the M point provide the possibility of additional tunability in this material[40,41] as electrons can propagate differently from one direction to another. The calculated

bands along the K–M–K direction show excellent agreement with our experimentally measured dispersion maps (see Fig. 2(d)). We find that the energy window of linear dispersion is more than 2.3 eV, extending both below and above the Fermi level, which is more than any known topological material. Our calculations reveal an even number of topological nontrivial surface states, a Dirac cone at $\Gamma$ and one at M, and the Dirac-node arc along the M–$\Gamma$ direction, all due to the bulk band set (A, B)—a clear signature of a weak $\mathbb{Z}_2$ invariant. Furthermore, besides the strong and weak TI states above the chemical potential, we also observe a Dirac-like state at the same momentum position, $\Gamma$, at about $E_B = 1.2$ eV, consistent with the ARPES observation (see Supplementary Fig. 2). However, the topological origin of this state is unclear.

In total, four surface Dirac states are reported here at TRIMs. Two of them, one at the M point around 0.9 eV below the Fermi level and one at the $\Gamma$ point around 0.17 eV above the Fermi level, stem from bands A and B and have weak $\mathbb{Z}_2$ topological origin (green box in Fig. 4(b)). The Dirac state at the $\Gamma$ point around 0.42 eV above the Fermi level stems from bands B and C and has a strong $\mathbb{Z}_2$ topological origin (purple box in Fig. 4(b)). Lastly, the Dirac point at $\Gamma$ around 1.2 eV below the Fermi level (black box in Fig. 4(b)) is unrelated to either bands A, B, or C and is included here for the sake of completeness. The Dirac states below the Fermi level are directly observed while the ones above the Fermi level are indirectly witnessed from their observed signatures at negative energies in our ARPES measurements.

## Discussion

Firstly, we observe a sixfold flower petal-shaped FS which shows that even in a metallic Dirac material such a remarkable dispersion is possible. Secondly, we observe multiple Dirac cones with linear dispersion over a wide energy range (~2.3 eV), even larger than that of ZrSiS (~2 eV). Most importantly, in the well-studied

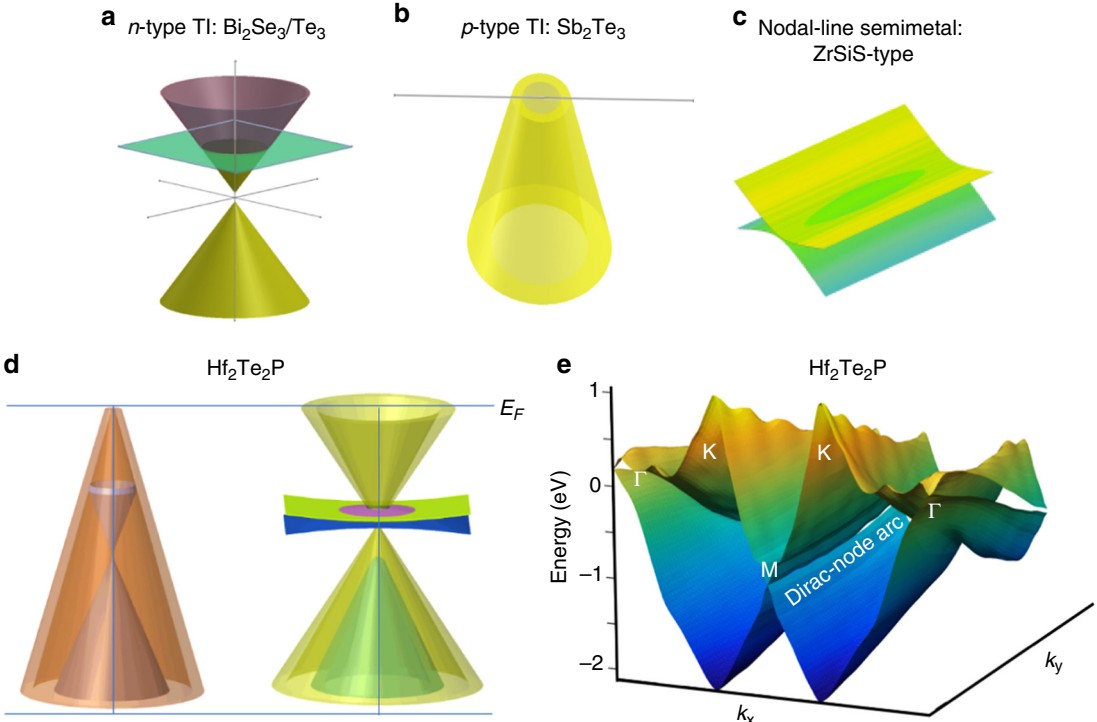

**Fig. 5** Schematic view of distinct fermionic states. **a**, **b** Sketch of electronic surface state dispersion of the *n*-type topological insulator and *p*-type topological insulator, respectively. **c** Dispersion of bulk states in a nodal-line semimetal of ZrSiS-type. **d** Sketch of electronic dispersions of the 221-material $Hf_2Te_2P$. This material consists of both *n*- and *p*-type topological surface states as well as a surface Dirac-node arc phase. **e** View of the calculated surface electronic structure of this material that confirms its weak topological nature, also showing the Dirac-node arc starting at M. The Dirac-node arc is purely surface-derived, in contrast to the nodal-line semimetal phase shown in **c**, which is bulk-derived

typical *n*-type $Bi_2Se_3/Bi_2Te_3$[1,2] TI materials (Fig. 5(a)), it is experimentally observed that the surface Dirac cones have lower and upper cone giving the Fermi level well above the Dirac point. On the other hand, for the distinct *p*-type material such as $Sb_2Te_3$[1,2], the Dirac point is located well above the Fermi level (Fig. 5(b)). Furthermore, in ZrSiS-type nodal line materials, the bulk conduction and valence band touch each other along a 1D loop or line protected by nonsymmorphic symmetry[26,27] (Fig. 5 (c)). Notably, these three phenomena are uniquely found in a distinct material or family of materials. In contrast, here, we have acquired sufficient experimental and theoretical evidence to report that $Hf_2Te_2P$ hosts multiple surface Dirac states (Fig. 5(d), (e)). To our knowledge, never before has a single material been found to host three such topological states.

The Dirac-node arc along Γ–M that originates from the topological bulk band set (A, B) is protected by the in-plane TRS —a 2D analog of the conventional TRS[33]. The {110}, {010}, and {100} mirror planes in the conventional unit cell of the $Hf_2Te_2P$ crystal suggest the presence of in-plane TRS and according to ref.[33], one finds topologically protected Dirac lines on the surface of weak TIs, here it is the (001) surface.

We analyze here that such line of (ideally) ungapped Dirac crossings emerges as topological edge states of sufficiently weakly coupled 2D planes stacked perpendicular to an axis that is tilted with respect to the direction of the line due to the presence of in-plane TRS in those planes[33]. For $Hf_2Te_2P$, the combination of, e.g., the {100} mirror symmetry and usual TRS ensures the existence of planes stacked parallel to the (100) direction that possess an in-plane TRS, thus giving rise to our observed surface Dirac-node arc along the (010) (Γ–M direction). Furthermore, at the surface where TRS is broken, the {100} mirror symmetry remains, thus protecting the Dirac node-arc along this direction. In a similar way, the combination of usual TRS and the {010} mirror symmetry gives rise

and protects the Dirac node-arc along the (100) direction. The rest of the Dirac node-arcs of the surface BZ can then be understood due to threefold rotational invariance. A plausible explanation why this line develops a gap as we move closer to Γ along the Γ–M high-symmetry direction of the surface BZ is that the interlayer coupling between these planes in the bulk is stronger around the Γ point, therefore the in-plane TRS is broken there. This picture is supported by the fact that the bulk electron bands A and B are highly dispersive near the Γ point in comparison to their almost flat shape near the F point, as shown in Fig. 4(a).

It is worth mentioning that our DFT calculations show the emergence of a small gap of a few meV in the Dirac dispersions in the region away from the M point, which nevertheless lies beneath the scale of our experimental resolution. We attribute the emergence of such tiny gap values to a very weak but finite interlayer coupling whose existence is unavoidable in a real material. In the limit of vanishing interlayer coupling, the Dirac-node arc would be symmetry protected. In this sense, the in-plane TRS that we discuss here is rather an approximate symmetry which, however, allows for the appearance of our observed surface node arc due to the above analysis. When we move far away from the M point along the M–Γ direction, the gap increases; we calculate a 30–40 meV gap which is visible in Fig. 5(e) and also experimentally resolvable as shown in Fig. 3(c).

We note that the Dirac-node arc has one endpoint at M and another one between M and Γ. For ideal 2D symmetry the latter endpoint would be the gapless weak topological Dirac surface states at the Γ point that exist at around 0.2 eV (see Fig. 5(e)). Therefore, our Dirac-node arc does not disperse into the bulk, and it is also distinct from the Fermi arc of Weyl semimetals. We furthermore note that the here-discovered surface Dirac-node arc is distinct as well from the Dirac electron state observed on the surface of $Ru_2Sn_3$ (ref. [42]) which consists of a single Dirac

crossing at the Γ point and a gapped surface state at momenta along a high-symmetry line. The Dirac-node arc in $Hf_2Te_2P$ consists of (approximately ungapped) Dirac crossings along a line in momentum space.

Interestingly, our calculations show that one of the Dirac nodes at the Γ point is a weak topological state with $\mathbb{Z}_2$ invariant (0:111) and that the other one is a strong topological state with $\mathbb{Z}_2$ invariant (1:111). In contrast to the other known topological materials, our first-principles calculations show that the band inversion is $d-p$ type similar with $Zr_2Te_2P$[31] instead of a $s-p$-type band inversion. The bands around the M points are highly anisotropic so there is a large possibility of a tunable Dirac cone state that may lead to new properties. This system provides a unique opportunity to study multiple fundamental fermionic quantum phases in the same material.

## Methods

**Crystal growth**. The single crystals of $Hf_2Te_2P$ were grown through the vapor transport method as described elsewhere[35]. The crystal structure was determined by X-ray diffraction on a Kuma-Diffraction KM4 four-circle diffractometer equipped with a CCD camera using Mo Kα radiation, while chemical composition was checked by energy dispersive X-ray analysis performed using a FEI scanning electron microscope equipped with an EDAX Genesis XM4 spectrometer. Electrical resistivity measurements were carried out within the temperature range 2–300 K and in applied magnetic fields of up to 9 T using a conventional four-point ac technique implemented in a Quantum Design on a four-circle PPMS platform. The electrical contacts were made using silver epoxy paste.

**Spectroscopic characterization**. Synchrotron-based ARPES measurements were performed at the SIS-HRPES end-station at the Swiss Light Source (SLS) equipped with Scienta R4000, Advanced Light Source (ALS) beamline 10.0.1 equipped with Scienta R4000 and ALS beamline 4.0.3 equipped with R8000 hemispherical electron analyzers. The angular resolution was set to be better than 0.2°, and the energy resolution was set to be better than 20 meV for the measurements. The samples were cleaved in situ under vacuum condition better than $3 \times 10^{-11}$ torr and at a temperature around 18 K.

**Electronic structure calculations**. The electronic structure calculations and structural optimization were carried out within the density-functional formalism as implemented in the Vienna ab initio simulation package (VASP)[43,44]. Exchange and correlation were treated within the generalized gradient approximation (GGA) using the parametrization of Perdew, Burke, and Ernzerhof (PBE)[45]. The projector-augmented wave (PAW) method[46,47] was employed for the wave functions and pseudopotentials to describe the interaction between the ion cores and valence electrons. The lattice constants and atomic geometries were fully optimized and obtained by minimization of the total energy of the bulk system. The surface of the 2D crystals was simulated as a slab calculation within the supercell approach with sufficiently thick vacuum layers. In addition to the general scalar-relativistic corrections in the Hamiltonian, the spin-orbit interaction was taken into account. The plane wave cutoff energy and the $k$-point sampling in the BZ integration were checked carefully to assure the numerical convergence of self-consistently determined quantities.

**Data availability**. All relevant data that support the findings of this study are available upon reasonable request to one of the corresponding authors.

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

## Acknowledgements

M.N. is supported by the Air Force Office of Scientific Research under Award No. FA9550-17-1-0415 and the startup fund from UCF. T.D. is supported by the NSF IR/D program. A.K.N., A.A., P.M., and P.M.O. acknowledge support from the Swedish Research Council (VR), the K. and A. Wallenberg Foundation (Grant No. 2015.0060), and the Swedish National Infrastructure for Computing (SNIC). We thank Nicholas Clark Plumb for beamline assistance at the SLS, PSI. We also thank Jonathan Denlinger and Sung-Kwan Mo for beamline assistance at the LBNL.

## Author contributions

M.N. conceived the study; D.K. synthesized the samples and performed the electrical transport characterization; M.M.H. performed the measurements with the help of K.D., G.D., F.K., C.S., T.D., R.S., F.C., and M.N.; M.M.H. and M.N. performed the data analysis and figure planning; A.K.N., A.A., P.M., and P.M.O. performed the ab initio calculations and topological analysis; P.M.O. was responsible for the theoretical research direction; M.M.H., A.K.N., A.A., P.M.O., and M.N. wrote the manuscript with input from all authors. M.N. was responsible for the overall research direction, planning, and integration among different research units.

## Additional information

**Competing interests:** The authors declare no competing interests.

