## [Peer Review File · Nature Communications]

Reviewers' comments:

Reviewer #2 (Remarks to the Author):

Hosen and co-authors present an ARPES and ab initio calculation study of Hf₂Te₂P. They observe multiple Dirac point nodes and a Dirac node arc, and a combination of both theory and experiment is used to argue for the coexistence of both strong and weak topological surface states. This work is an important development in the field. The richness of topological surface states observed and predicted in this one material, as well as the capability of this material to yield very sharp ARPES spectra, will prompt many followup studies. I feel that the authors have done an adequate job responding to my earlier referee report, and I recommend this paper for publication in Nature Communications.

A few remaining comments:

- Supplementary figure 2d: dirac crossing indicated in box is not at all clear
- Supplementary figure 8: MDC row is a bit hard to parse; could be large or with fewer MDCs focusing only near Dirac point
- Fig 5 is not particularly illustrative. The introduction and discussion do a better work of laying out the framework than this figure.

Reviewer #3 (Remarks to the Author):

The authors have improved and clarified their manuscript in response to the referee reports. However, the definition and explanation of the Dirac arc is still not clear. Since it is one of the main observations of the paper, it must be completely clear. I can envision two possibilities:

1. The Dirac arc is actually a Dirac point with a near-zero dispersion in one direction. This requires no symmetry. This has been observed in Ru₂Sn₃ (see Gibson, et al, Scientific Reports vol. 4, Article number: 5168 (2014)).
2. The Dirac arc is a two-fold degenerate set of bands that remain two-fold degenerate along some line/path in the surface Brillouin zone. The latter possibility requires symmetry to protect, which is referred to as "in-plane time reversal symmetry" in Ref. 33. The authors refer to this symmetry but do not explain why their system possesses it. I believe this could arise the product of time reversal and a mirror symmetry, but if the authors rely on this symmetry, the need to explain what it is and why their system has it.

The authors must clarify which of these possibilities (or perhaps something else) describes their Dirac arc.

I would like to reiterate from my previous report that the experimental data and DFT results are very nice.

Authors' response to reviewers' comments:

Reviewer 2:

Hosen and co-authors present an ARPES and ab initio calculation study of Hf₂Te₂P. They observe multiple Dirac point nodes and a Dirac node arc, and a combination of both theory and experiment is used to argue for the coexistence of both strong and weak topological surface states. This work is an important development in the field. The richness of topological surface states observed and predicted in this one material, as well as the capability of this material to yield very sharp ARPES spectra, will prompt many followup studies. I feel that the authors have done an adequate job responding to my earlier referee report, and I recommend this paper for publication in Nature Communications.

Authors: We would like to thank the reviewer for his/her time and recommending our manuscript for publication in *Nature Communications*.

A few remaining comments:

-Supplementary figure 2d: dirac crossing indicated in box is not at all clear

Authors: We wish to thank the reviewer for his/her comments. We have created a new supplementary figure 2 focusing near the Dirac point. The new Figure S 2d will help to better observe the Dirac crossing. However, we would like to note that, Dirac dispersion here is buried on the bulk band which makes it less visible. We agree with the reviewer that Dirac crossing is not clear due to the presence of other irrelevant bands. The new figure looks as follows:

-Supplementary figure 8: MDC row is a bit hard to parse; could be large or with fewer MDCs focusing only near Dirac point

Authors: We wish to thank the reviewer for this suggestion. We have inserted new MDC row with fewer MDCs to clearly see the Dirac point. The new figure looks as follows:

-Fig 5 is not particularly illustrative. The introduction and discussion do a better work of laying out the framework than this figure.

Authors: We wish to thank the reviewer for this comment. We have modified our Fig. 5 to represent our data better. We respect the reviewer's opinion; however, we think that it might help the reader in some extent to visualize our data and wish to keep this figure. The new Fig. 5 looks as follows:

Reviewer 3:

The authors have improved and clarified their manuscript in response to the referee reports. However, the definition and explanation of the Dirac arc is still not clear. Since it is one of the main observations of the paper, it must be completely clear. I can envision two possibilities:

1. The Dirac arc is actually a Dirac point with a near-zero dispersion in one direction. This requires no symmetry. This has been observed in Ru₂Sn₃ (see Gibson, et al, Scientific Reports vol. 4, Article number: 5168 (2014)).

2. The Dirac arc is a two-fold degenerate set of bands that remain two-fold degenerate along some line/path in the surface Brillouin zone. The latter possibility requires symmetry to protect, which is referred to as "in-plane time reversal symmetry" in Ref. 33. The authors refer to this symmetry but do not explain why their system possesses it. I believe this could arise the product of time reversal and a mirror symmetry, but if the authors rely on this symmetry, the need to explain what it is and why their system has it.

The authors must clarify which of these possibilities (or perhaps something else) describes their Dirac arc.

Authors: We thank the reviewer for appreciating our efforts to improve the manuscript. To address the point of the reviewer regarding the origin of the observed Dirac node arc we have added a further text part in the Discussion section. First, in the introduction a few words were added to explain that the Dirac node arc is a line segment along which there is a Dirac crossing in one direction. Second, the Dirac arc that we discuss is different from the situation discussed for Ru₂Sn₃ by Gibson *et al* (Sci. Rep. **4**, 5168 (2014)). In that paper there is a single Dirac crossing at the Gamma point and a gapped surface state at other momenta along a high-symmetry line. Our observed Dirac node arc consists of ungapped Dirac crossings along a line in momentum space. Moreover, neither the actual Dirac crossing is observed (as it is above EF) nor the top of the gapped of the gapped Dirac cones. In our work we observe both experimentally and theoretically gapless Dirac crossings along a line in the surface Brillouin zone.

The situation at hand in Hf₂Te₂P is more closely described by the 2nd possibility mentioned by the reviewer. We have added a further analysis to the Discussion section, which follows up on this aspect. We had already mentioned previously the "in-plane time-reversal symmetry" in relation to the work of Ref. 33; in the revised version we now clarify its origin in more detail.

I would like to reiterate from my previous report that the experimental data and DFT results are very nice.

Authors: We would like to thank the reviewer for finding our experimental data and calculated results very nice.

REVIEWERS' COMMENTS:

Reviewer #3 (Remarks to the Author):

The authors argue that their Dirac node arc is a result of "in-plane time-reversal symmetry," as defined in Ref [33]. However, they do not provide a reason why this symmetry would be robust. In fact, as the authors describe, the data in Fig 3 showing that the two-fold degeneracy does not span the entire line from Gamma-M proves that the symmetry is not robust. Thus, I believe the Dirac node arc is not an exact two-fold degeneracy, but a result of an approximate symmetry, which, as the authors explain in the revised Discussion, can result from weakly coupled planes with a nontrivial Z_2 invariant (consistent with the weak TI indices.) In this case, the name Dirac node arc is slightly misleading since the Dirac degeneracy is not exact along the arc.

Further evidence for/against this theory could be obtained by zooming in further in Fig 5e to check whether the Dirac degeneracy along the arc survives or becomes gapped at energies beneath the experimental measurement scale.

I believe the paper is suitable for publication after the authors clarify the point that the symmetry is approximate and thus the apparent Dirac degeneracy line is actually gapped (beneath the scale of the experiment). I like the new text in the Discussion that posits an explanation for why the approximate symmetry is more robust near Gamma than near M.

But, as it stands, I think the analysis is misleading unless the authors can provide further evidence for why there would be a robust in-plane time-reversal symmetry that disappears partway along the Gamma-M line.

Authors' response to reviewers' comments:

The authors argue that their Dirac node arc is a result of "in-plane time-reversal symmetry," as defined in Ref [33]. However, they do not provide a reason why this symmetry would be robust. In fact, as the authors describe, the data in Fig 3 showing that the two-fold degeneracy does not span the entire line from Gamma-M proves that the symmetry is not robust. Thus, I believe the Dirac node arc is not an exact two-fold degeneracy, but a result of an approximate symmetry, which, as the authors explain in the revised Discussion, can result from weakly coupled planes with a nontrivial Z₂ invariant (consistent with the weak TI indices.) In this case, the name Dirac node arc is slightly misleading since the Dirac degeneracy is not exact along the arc.

Further evidence for/against this theory could be obtained by zooming in further in Fig 5e to check whether the Dirac degeneracy along the arc survives or becomes gapped at energies beneath the experimental measurement scale.

I believe the paper is suitable for publication after the authors clarify the point that the symmetry is approximate and thus the apparent Dirac degeneracy line is actually gapped (beneath the scale of the experiment). I like the new text in the Discussion that posits an explanation for why the approximate symmetry is more robust near Gamma than near M.

But, as it stands, I think the analysis is misleading unless the authors can provide further evidence for why there would be a robust in-plane time-reversal symmetry that disappears partway along the Gamma-M line.

Authors: We would like to thank the reviewer once more for the important comments and constructive criticism. We agree that the in-plane TRS is an approximate symmetry, depending on the weak but nonzero interlayer coupling, and have added a paragraph in the main text to address this point in more detail (see page 4, last Paragraph). We also clarify in the main text that there is indeed a tiny gap, as calculated within DFT, farther away from the M point along the Gamma-M line which is nevertheless below the experimental resolution. We hope that our revised manuscript is now suitable for publication in *Nature Communications*.